# Cellulose $\delta^{18}$O of Tree Rings Reflects Vapour Pressure Variations in the Ordos Plateau

**Wentai Liu** [1,2], **Qiang Li** [1,3,4,*], **Huiming Song** [1,3,4], **Ruolan Deng** [5] **and Yu Liu** [1,3,4,6]

1   State Key Laboratory of Loess and Quaternary Geology, Institute of Earth Environment, Chinese Academy of Sciences, Xi'an 710061, China; liuwentai@ieecas.cn (W.L.); songhm@ieecas.cn (H.S.); liuyu@loess.llqg.ac.cn (Y.L.)
2   College of Earth and Planetary Sciences, University of the Chinese Academy of Sciences, Beijing 100049, China
3   CAS Center for Excellence in Quaternary Science and Global Change, Xi'an 710061, China
4   China-Pakistan Joint Research Center on Earth Sciences, CAS-HEC, Islamabad 45320, Pakistan
5   Key Laboratory of Subsurface Hydrology and Ecological Effects in Arid Region, School of Water and Environment, Chang'an University, Xi'an 710054, China; dengruolan12@163.com
6   School of Human Settlements and Civil Engineering, Xi'an Jiaotong University, Xi'an 710049, China
*   Correspondence: liqiang@ieecas.cn; Tel.: +86-29-6233-6226

**Abstract:** In arid and semi-arid regions, a better understanding of the effect of climate change mechanisms on environmental evolution can be used to guide regional ecological conservation and to improve water resource availability. Increased aridity in arid and semi-arid regions considerably affects the physiological functions of plants and the exchange of carbon and water with the environment. We collected *Pinus tabuliformis* Carr. samples from Ordos, Inner Mongolia, and measured their $\delta^{18}$O variations. Vapour pressure (VP) was the main factor dominating $\delta^{18}$O variations from July to August, indicating the regulatory role of plant leaf stomata. Based on the $\delta^{18}$O series in the Ordos region, we reconstructed VP variations for July–August ($VP_{JA}$) for the past 205 years. Spatial analysis showed the reconstruction as spatially highly representative. VP variations in the Ordos region mainly reflected precipitation variations and did not show a significant correlation with temperature. Since the late 1950s, VP has been decreasing, which is related to the weakening of the Asian monsoon. The results of reconstruction decomposed using ensemble empirical mode decomposition showed that El Niño–Southern Oscillation may affect VP in the study area, and the effect of sea surface temperature on the central and eastern Pacific Ocean in the Ordos region may lead to an increase in the drought.

**Keywords:** tree-ring $\delta^{18}$O; vapour pressure reconstruction; Ordos Plateau; Asian monsoon

## 1. Introduction

During 1951–2012, global average annual temperature showed an increasing trend of about 0.12 °C per decade [1]. In the future, continued warming can lead to an increase in the atmospheric water demand. Plant leaves volatilise soil water into the atmosphere through transpiration, thereby increasing soil aridification and water loss from forest systems. Vapour pressure (VP) plays a critical role in the transpiration process; thus, understanding long-term VP changes can increase our understanding of the correlation between the water cycle within species and climate change. The short duration of instrumental observations of VP often limits our understanding of VP changes for contemporary global climate changes. Therefore, on long-time scales, VP change reconstruction using paleoclimate proxies is highly beneficial for comprehending both the present-day global climate changes and future trends.

Tree rings are considered a suitable proxy for recording climate changes [2]. Most studies have been conducted based on tree-ring width. Reconstruction based on tree-ring width/density presents a potential deficit in low-frequency signals, which is a well-known

fatal problem because long-term growth trends must be eliminated using treatments (e.g., negative exponential functions) [3,4]. A stable oxygen isotope ratio ($\delta^{18}$O) in plant cellulose leads to the good preservation of paleoclimate signals [5,6]. Stable isotopes in tree cellulose are physically more explanatory and correlate more strongly with climatic factors than tree-ring width/density indices. Stable oxygen isotope measurements do not require mathematical modelling, can be directly used for climate analyses, and can result in satisfactory preservation of low- and high-frequency climate signals. Therefore, tree-ring stable isotopes are increasingly becoming the preferred proxy for paleoclimate variation reconstruction [7].

Tree-ring cellulose $\delta^{18}$O is a combination of three factors: $\delta^{18}$O in source water, leaf evaporative enrichment, and chemical fractionation in plant organic matter syntheses [6,8,9]. $\delta^{18}$O in source water, which is mainly derived from precipitation, is the main source of $\delta^{18}$O in cellulose [8,10]. Due to the strong correlation between precipitation $\delta^{18}$O and climate change [11], plant cellulose $\delta^{18}$O is considered a proxy for the reconstruction of past climate and atmospheric circulation changes [7,12–16]. During transpiration, the evaporative loss of $H_2^{16}$O from leaves is higher than that of $H_2^{18}$O [8]; hence, the enrichment of leaf water $\delta^{18}$O may increase under drought conditions. At constant temperature, enrichment resulting from evaporation is controlled through ambient pressure and intercellular VP [17]. Thus, a mixture of environmental and physiological factors influences plant cellulose $\delta^{18}$O.

Correlation analyses provide the strong and weak relationships between climatic variables (e.g., temperature, precipitation, relative humidity and VP) and cellulose $\delta^{18}$O [12,14,18,19]. At mid and high latitudes, precipitation $\delta^{18}$O strongly depends on temperature [11,20]. Li et al. (2020) were the first researchers to reconstruct 1818–2012 temperature variations in north-eastern China based on tree-ring $\delta^{18}$O [21]. Relative humidity is an indicator describing the water vapour content of the air, which can affect transpiration in plants, and thus, oxygen isotope fractionation in leaf water. Under dry and warm (low relative humidity) conditions, VP in air decreases and $^{18}$O enrichment is high, which renders tree-ring $\delta^{18}$O to be negatively correlated to relative humidity and VP [8,12,14,15,18]. Transpiration $^{18}$O enrichment processes influenced by VP changes may affect temperature signals in tree rings, making the detection of these rings considerably difficult. In the absence of other environmental information, whether temperature dominates cellulose $\delta^{18}$O cannot be determined [22]. When temperature increases due to global warming, VP in air decreases and evapotranspiration increases [23], and this change may alter $^{18}$O fractionation in tree water. Therefore, improving the understanding of long-term changes in atmospheric VP by establishing tree ring $\delta^{18}$O data for the past several centuries is important.

The Ordos Plateau is in the transition zone between arid and semi-arid regions, with an annual precipitation of 200–400 mm, and is sensitive to changes in the Asian summer monsoon [24]. In this study, we used the tree-ring $\delta^{18}$O of *Pinus tabuliformis* Carr. growing in the Ordos Plateau to reconstruct VP changes since 1808, based on statistically significant relationships and physiological mechanisms. We hypothesised that an increase in temperatures in the study area led to an increase in transpiration, water demand for plant transpiration, and stomatal water loss and to higher $\delta^{18}$O values in trees. Trees with considerable stomatal changes in response to climate changes have water-related $\delta^{18}$O that masks the temperature change signal, which rendered our VP reconstruction feasible.

## 2. Materials and Methods

### 2.1. Study Area and Sampling

The study was conducted in Ordos, the south-western Inner Mongolia Autonomous Region, China (Figure 1a), located in the hinterland of the Ordos Plateau at an altitude of approximately 1200–2900 m. The study area is located in the northern temperate semi-arid continental climate zone. According to the meteorological records of the Dongsheng meteorological station (39°50N, 109°59′E, 1157.0 m a.s.l.), for 1957–2012, the annual average

temperature, annual average precipitation, annual average relative humidity and annual average VP were 6.2 °C, 386 mm, 49.4% and 5.9 hpa, respectively. The lowest and highest temperatures are attained in January and July, with monthly average temperatures of −10.7 °C and 21.2 °C, respectively. Between June and August, 64% of the annual precipitation is concentrated, with the driest and wettest months of April and August, respectively, with the monthly average relative humidity of 36.3% and 63.2%, respectively. The months with high VP are July and August, with the monthly average VP of 13.7 and 13.5 hpa, respectively (Figure 1b,c).

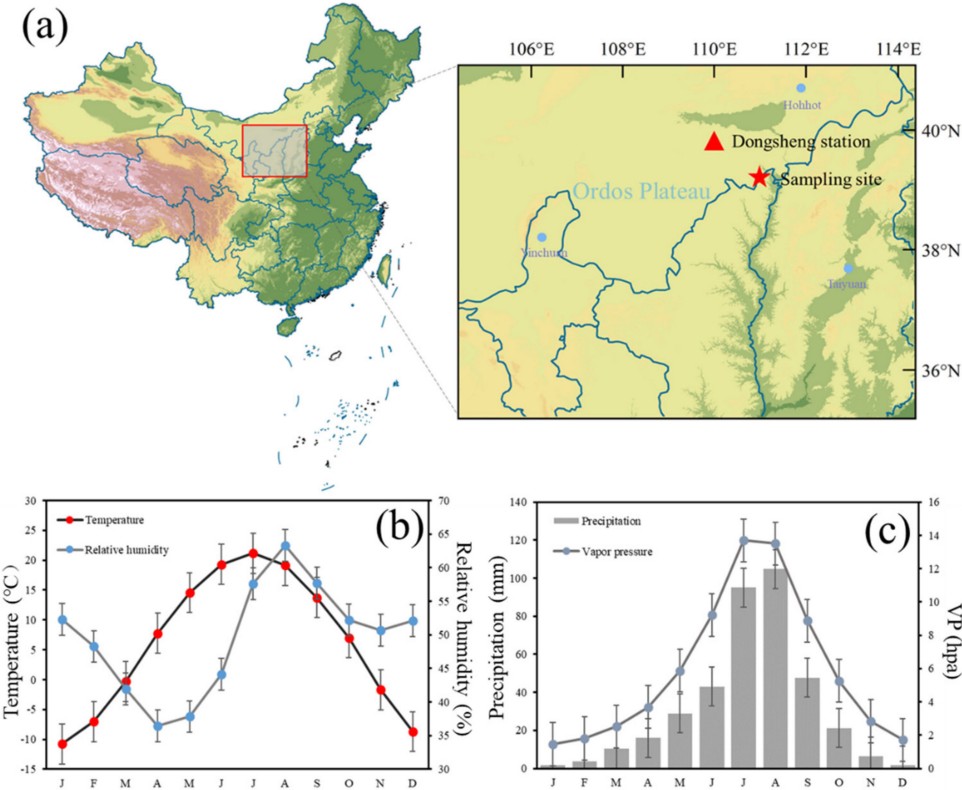

**Figure 1.** (**a**) Location of the study site and the nearest meteorological station (Dongsheng station). (**b**) Monthly distributions of temperature and relative humidity and (**c**) precipitation and VP at Dongsheng meteorological station from 1957 to 2012.

In July 2013, we collected a set of *P. tabulaeformis* Carr. cores (almost two cores per tree, at right angles, collected from approximately 1.3 m above the ground) by using a 5-mm diameter incremental auger. The sampling sites were located on a flat, open plateau, with the pines located 1–3 km apart and unobstructed from each other.

## 2.2. Sample Preparation and Cellulose $\delta^{18}O$ Measurement

Tree-ring widths (accuracy 0.01 mm) were first measured sequentially from the bark to pith of the core by using a sliding scale micrometre and were recorded in the Lintab 6 system. Cross dating was then performed using COFECHA software.

Conducting stable isotope analyses representative of environmental conditions in the study area required the selection of four or five tree cores [25]. We selected four cores without significant differences in growth and with normal growth, no missing rings, and clear boundaries between tree rings for isotopic analyses. The average length of the four cores, namely ERDS1 (1808–2012 CE), ERDS2 (1871–2012 CE), ERDS3 (1811–2012 CE) and ERDS 4 (1829–2012 CE) was 183 years. α-Cellulose was extracted using the modified Jayme–Wise method [26]. Under a microscope, the earlywood and latewood were cut together to ensure a sufficient sample size for chemical reactions. Subsequently, each year's

sample was transferred to a test tube and was chemically treated to obtain α-cellulose [27]. Afterwards, cellulose was homogenised in a cell grinder, and water was removed using the freeze–drying method.

During measurements, we weighed the homogeneous dried α-cellulose samples (0.12–0.16 mg) in silver capsules. The isotopic ratios ($^{18}O/^{16}O$) were measured in the tree-ring laboratory of the Institute of Earth Environment, the Chinese Academy of Sciences, by using a high-temperature transform elemental analyser (TC/EA) at high temperature in combination with an isotope ratio mass spectrometer (Delta V Advantage, ThermoFisher). We used Merck microcrystalline cellulose (27.7‰) as the laboratory standard and inserted one standard for eight samples. The analytical uncertainty of $\delta^{18}O$ measured using MERCK cellulose was 0.2‰ (N = 91), which can be achieved to correct isotopic values and ensure measurement quality [21]. The $\delta^{18}O$ was calculated as:

$$\delta^{18}O = (R_{sample}/R_{standard} - 1) \times 1000 \tag{1}$$

where $R_{sample}$ and $R_{standard}$ are the $^{18}O/^{16}O$ ratios of the sample and standard, respectively.

### 2.3. Climate Data

The precipitation, monthly mean temperature, monthly mean relative humidity, and monthly mean VP data of three nearby meteorological stations (Dongsheng, Yulin and Baotou) were obtained from the China Meteorological Service Centre (http://data.cma.cn/, accessed on 12 September 2020). The data from the Dongsheng meteorological station, which is closest to the sampling site, were used for reconstruction, and an average of the data from the remaining two meteorological stations was used for comparison. We selected all the instrumentally observed data from 1957 to 2012.

### 2.4. Statistical Methods

A numerical mixing method was employed to generate a master series to represent $\delta^{18}O$ variations in the Ordos region [28]. The expressed population signals (EPSs) between individual tree cores over a 30-year time window were calculated using Equation (1):

$$EPS = (n \times Rbar)/[1 + (n - 1) \times Rbar] \tag{2}$$

where n is the number of time series, and Rbar is the mean inter-series correlation [29].

Pearson (r) and partial correlations were used to test the correlation between the $\delta^{18}O$ series and individual climate factors. The split method was employed to analyse the stability and reliability of the reconstruction equation during the calibration and verification of the linear regression model [30]. The statistical parameters used for validation included the correlation coefficient (r), variance ($R^2$), reduced error test (RE), coefficient of efficiency (CE) and product mean test (t).

We spatially correlated reconstruction with precipitation, PDSI and sea-level temperature data by using the Royal Netherlands Meteorological Institute Climate Explorer (http://www.knmi.nl/, accessed on 12 September 2020). Data were obtained from the Royal Netherlands Meteorological Institute Climate Explorer (http://climexp.knmi.nl/, accessed on 12 September 2020).

The reconstructed sequence was decomposed through the ensemble empirical modal decomposition (EEMD) into several sequences with different characteristics, called intrinsic mode functions (IMFs) [31].

## 3. Results

### 3.1. $\delta^{18}O$ Series in the Ordos Region

We used the numerical mixing method to compose the four tree-ring $\delta^{18}O$ sequences together to generate a master sequence covering the period of 1808–2012 in the Ordos region (Figure S1a). The Rbar and EPS of the four sequences over the common period are

0.43–0.65 and 0.73–0.88, respectively. Table S1 presents the statistical characteristics of the four independent sequences and main sequence, and Table S2 presents the correlations among the individual sequences, with the significant correlations of 0.527–0.707 between each two sequences. The kurtosis and skewness values illustrate a good structure of the master series with a high similarity to typical Gaussian distribution. The mean of the synthesised master series is 32.38‰, with the standard deviation, maximum value, and minimum value of 1.24, 35.47‰, and 29.28‰, respectively (Table S1) [32].

We obtained the $\delta^{18}O_p$ data from the Global Network of Isotopes in Precipitation project (http://www-naweb.iaea.org, accessed on 25 September 2020) for two sites near the study area, Xi'an, and Baotou, for comparison with the annual $\delta^{18}O$ of trees in Ordos (Figure S2). $\delta^{18}O$ measured was consistent with the general trend of precipitation $\delta^{18}O_p$ variations at both the sites. An increase in precipitation leads to more negative precipitation $\delta^{18}O$ and vice versa [33]. The comparison results are in agreement with stable isotope grading theory and confirm that annual tree $\delta^{18}O$ inherits $\delta^{18}O_p$ from atmospheric precipitation [32].

### 3.2. Climate Response

The Dongsheng weather station is approximately 79 km away from the sampling site (Figure 1a). The mean $\delta^{18}O$ series of cellulose was not significantly correlated with the temperature data of instrumental observations (1957–2012) except for January and May; however, it is significantly correlated with relative humidity in July and August ($r = 0.667$; $p < 0.01$) and with $VP_{JA}$ ($r = 0.688$; $p < 0.01$). For all the months except October, the correlation between $\delta^{18}O$ and precipitation does not exceed the 99% confidence level (Figure 2a). The correlation of $\delta^{18}O$ series with VP is stronger than that with relative humidity in July and August. The aforementioned results indicated that the response of $\delta^{18}O$ to relative humidity and VP is better for tree rings in the Ordos region than that to precipitation and temperature. VP in the air reflects evapotranspiration in the environment, which affects the stomatal conductance of trees and determines the $\delta^{18}O$ fractionation rate [6]. In contrast to VP, temperature, precipitation and relative humidity all indirectly influenced $\delta^{18}O$ in tree rings by affecting $\delta^{18}O$ in precipitation, which may explain the strong correlation between tree-ring $\delta^{18}O$ and VP.

For verification, we calculated the partial correlation among annual tree-ring $\delta^{18}O$, temperature, precipitation, relative humidity and $VP_{JA}$. The correlation coefficients between VP and $\delta^{18}O$ were highest in July–August when temperature, precipitation, and relative humidity were the fixed variables (Table 1).

**Table 1.** Analysis of partial correlation between the tree ring $\delta^{18}O$ chronology and climatic records from July to August (1957–2012).

| Controlled Variable | δ18O vs. $T_{JA}$ | δ18O vs. $P_{JA}$ | δ18O vs. $RH_{JA}$ | δ18O vs. $VP_{JA}$ |
|---|---|---|---|---|
| $T_{JA}$ | | −0.287 | −0.655 * | −0.666 * |
| $P_{JA}$ | 0.192 | | −0.599 * | −0.619 * |
| $RH_{JA}$ | −0.280 | 0.122 | | −0.312 |
| $VP_{JA}$ | 0.234 | 0.035 | −0.222 | |

Note. $T_{JA}$, $RH_{JA}$, and $VP_{JA}$ are the mean July–August temperature's mean relative humidity, and mean VP, respectively. $P_{JA}$ is the total precipitation from July to August. * Significant at the 99% confidence level.

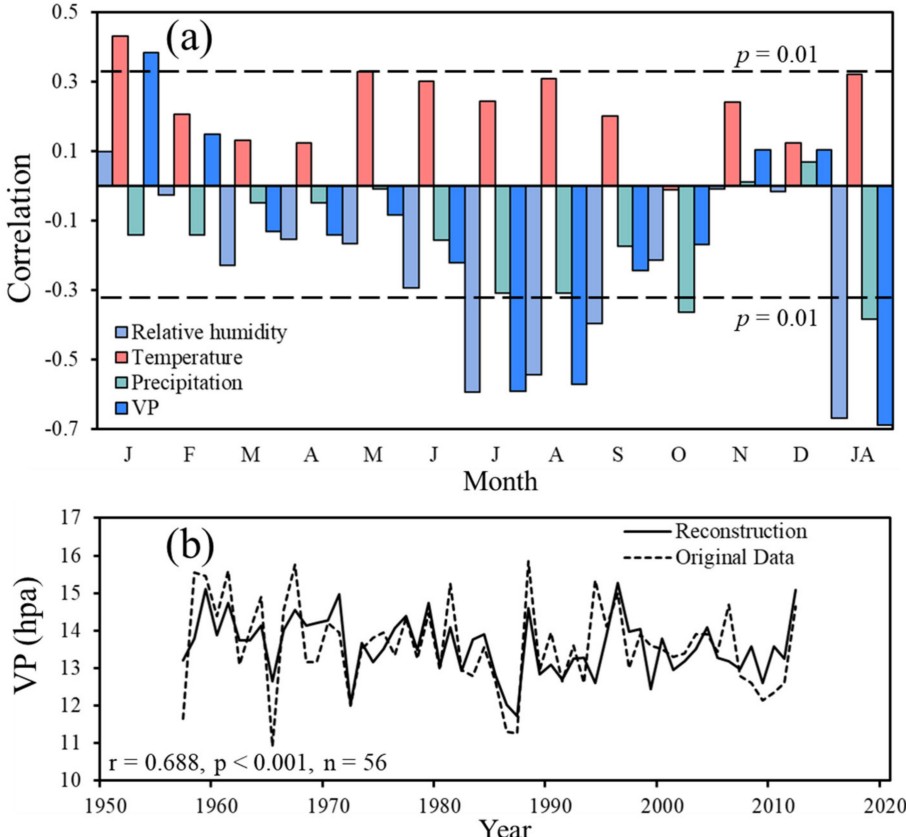

**Figure 2.** (**a**) Correlations between tree-ring $\delta^{18}O$ chronology and climatic factors (including relative humidity, temperature, precipitation, and VP). JA represents the period from July to August. (**b**) Comparison between reconstructed and observed July to August VP.

### 3.3. VP Reconstruction

We reconstructed $VP_{JA}$ for the growing season (July–August) based on linear regression models for $\delta^{18}O$ and VP datasets from 1957 to 2012. The linear regression function we designed is shown below:

$$VP_{JA} = 39.211 - 0.786 \times \delta^{18}O$$
$$(N = 56, r = -0.688, R^2 = 0.473, R^2_{adj} = 0.463, F = 48.426) \tag{3}$$

The reconstructed $VP_{JA}$ and actual instrumental observations showed highly coherent variations. A strong relationship existed over a common period of 56 years (r = 0.688, $p < 0.001$, N = 56) (Figure 2b). We assessed the model validity for the periods 1957–1986 and 1983–2012 by using standard segmentation tests (Table 2). Two rigorous tests with positive values for error (RE and CE) validated the regression model and the applicability of the time series to the climate reconstruction, respectively. The $VP_{JA}$ results reconstructed for 1808–2012 (Figure 3) showed a decreasing trend in three time intervals, 1819–1860, 1872–1928, and 1954–1987, which indicated climate changes from relatively wet to dry for these three intervals.

**Table 2.** Statistics of calibration verification tests for $VP_{JA}$ reconstruction in Ordos.

| Calibration | | | | | Verification | | | | | | |
|---|---|---|---|---|---|---|---|---|---|---|---|
| Period | r | $R^2$ | ST | t | Period | r | $R^2$ | RE | CE | ST | t |
| 1957–1986 | 0.772 | 0.596 | 23 | 4.438 | 1987–2012 | 0.587 | 0.345 | 0.171 | 0.146 | 18 | 3.565 |
| 1983–2012 | 0.626 | 0.392 | 19 | 3.981 | 1957–1982 | 0.730 | 0.533 | 0.545 | 0.475 | 19 | 4.013 |
| 1957–2012 | 0.688 | 0.473 | 39 | 5.911 | | | | | | | |

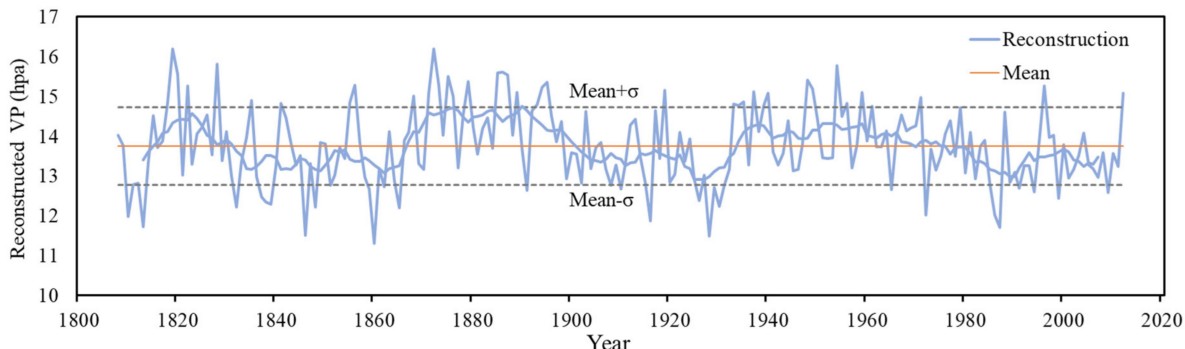

**Figure 3.** $VP_{JA}$ reconstruction during 1808–2012 in Ordos. The thick line is the 11-year moving averaged annual $VP_{JA}$.

We used KNMI climatic explorer software (http://www.knmi.nl, accessed on 12 September 2020) to analyse the spatial correlation between precipitation and reconstructed $VP_{JA}$ and between precipitation and the mean values of $VP_{JA}$ observed at the two meteorological stations near the sampling site (Yulin and Baotou meteorological stations) for the period 1957–2012 (Figure 4a,b). The spatial correlation analysis showed that reconstructed $VP_{JA}$ and precipitation based on the meteorological data acquired from 1957 to 2012 instrumental observations are significantly positively correlated and are highly spatially representative over a large area (Figure 4a). According to the comparison of the spatial correlation of the two meteorological station data (Figure 4b), although our reconstructed results did not show significant correlation on a larger scale, they exhibited the highest correlation with precipitation in the study area recorded during 1957–2012.

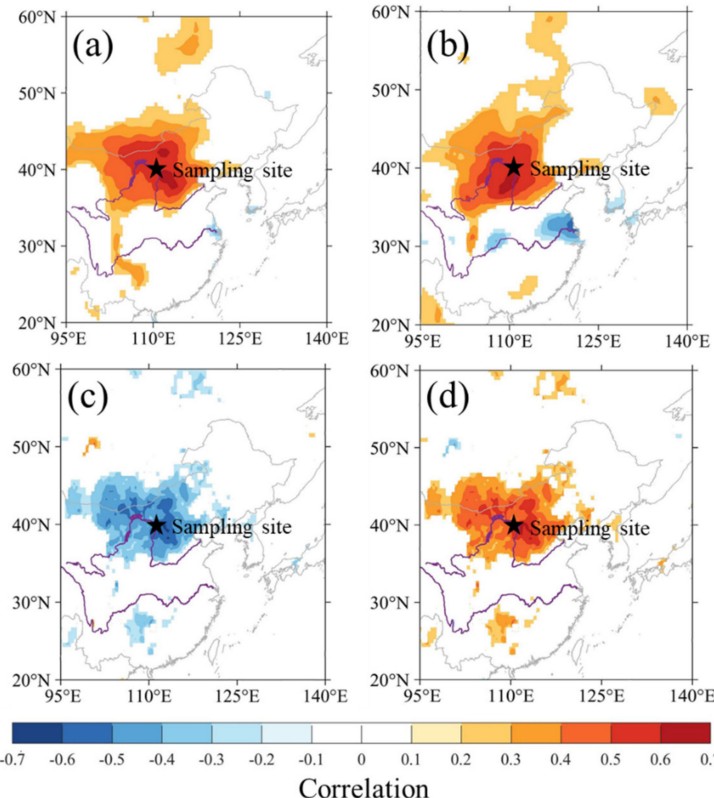

**Figure 4.** (**a**) Spatial correlation between precipitation and reconstruction $VP_{JA}$ during 1957–2012. (**b**) Spatial correlation between precipitation and the mean values of $VP_{JA}$ observed at two meteorological stations near the sampling site (Yulin and Baotou meteorological stations) during 1957–2012. (**c**) Spatial correlation between tree ring $\delta^{18}O$ and the PDSI drought index during 1901–2012. (**d**) Spatial correlation between reconstruction $VP_{JA}$ and the PDSI drought index during 1901–2012.

Based on the reconstruction of VP$_{JA}$ for 1808–2012, no clear long-term trend was observed on the centennial scale (Figure 3). The two periods with high VP were 1819–1828 and 1872–1896, and the lowest VP occurred in 1860.

## 4. Discussion

### 4.1. Climate Significance of Tree-Ring $\delta^{18}O$

Because many variables affect precipitation $\delta^{18}O_p$ values [11,20,34,35], which signals from the environment were recorded in cellulose $\delta^{18}O$ is unclear. When the main source of plant water is precipitation, the $\delta^{18}O_p$ signal from precipitation can be directly transferred to the plant cellulose for recording. According to Johnson and Ingram (2004), who studied the spatial distribution of precipitation $\delta^{18}O_p$ in China and factors influencing it, temperature is the main controlling factor in the northern part of China (either northwest or northeast) [20]. Therefore, *Pinus tabuliformis* Carr. tree-ring $\delta^{18}O$ may partially reflect the influence of temperature variations. However, in this study, *Pinus tabuliformis* Carr. tree-ring $\delta^{18}O$ did not show a significant correlation with temperature (Figure 2a) perhaps because the enrichment of leaf water molecules in response to VP variations in leaves lead to a change in temperature signals. Approximately 42% oxygen atoms in conifer carbohydrates are exchanged with xylem sap and carry the signal along with them [8], which is a mechanism that results in the mixing of the effects of environmental water source $\delta^{18}O$ and leaf enrichment. For decreased VP and increased VP deficit due to a restricted water source, the $\delta^{18}O$ signal of the water source in the environment inherited by the $\delta^{18}O$ in tree rings decreases [36]. In the absence of other information, the temperature $\delta^{18}O$ signal associated with water cannot be separated.

The correlation between tree-ring $\delta^{18}O$ and relative humidity and between VP and precipitation is higher than that with temperature (Figure 2a), indicating that tree rings are more strongly correlated with moisture conditions. This finding suggested that in cold and arid regions, such as Ordos, VP variations in the surrounding environment play a more critical role in the degree of oxygen isotope fractionation in trees than temperature variations. Several studies have concluded that moisture conditions in the environment have a moderating effect on the annual $\delta^{18}O$ of trees [18,37]. To further verify this conclusion, we spatially correlated the annual ring $\delta^{18}O$ of Ordos trees with the PDSI drought index by using KNMI climatic explorer software (http://www.knmi.nl, accessed on 12 September 2020) (Figure 4c) and found that the tree-ring $\delta^{18}O$ exhibited a strong spatial correlation. We reconstructed VP$_{JA}$ variations for 1808–2012, and the calibration and validation statistics indicated that our reconstruction was reliable (Table 2).

### 4.2. VP Variations during the Past Two Centuries

To examine the spatial representativeness of Ordos reconstruction VP$_{JA}$, we analysed the reported correlation between our reconstruction and gridded precipitation and between VP and PDSI datasets for 1901–2012 (Figure 4d). The areas of significant spatial correlations include Inner Mongolia, Shanxi, Shaanxi, and Ningxia regions. Both our reconstructed results and previous reconstructed records of the Asian monsoon marginal region can represent the variability of the Asian summer monsoon to varying degrees [24,33,38]. The Indian and East Asian monsoons can independently or together affect climate change, and both these monsoons belong to the Asian monsoon system [38]. Liu et al. (2019) showed that during June–August, the reconstructed results of relative humidity in the Ordos region were significantly correlated with the representative mean indices of the Asian summer monsoon in the east and south [38]. Our VP$_{JA}$ reconstruction results were closely correlated with July–August precipitation (Figure 4a). Therefore, reconstructed VP$_{JA}$ should have some correlation with the Asian summer monsoon.

The past 205 years include five extremely dry years of 1813, 1846, 1860, 1928 and 1987 and four extremely wet years of 1819, 1828, 1872 and 1954. In the Ordos region, wet years occur more frequently than dry years. According to historical records, both drought and wet years are well recorded (Table 3). The variations in drought and wet

substantially influenced the agriculture and economy of the respective time. In the last two centuries, the driest year was 1860 with a $\delta^{18}$O and reconstructed VP of 35.47‰ and 11.32 hpa, respectively. The devastating drought of 1920 was recorded in both geological record proxies and historical documentary records [39–41]. According to the historical records, the Ordos region has suffered from severe droughts since 1924. The drought of 1926–1929 was the most severe. The land was infertile and could not produce enough food due to pests and diseases, bandits were rampant, people lived on bark and roots, and the number of people affected was 40,000 [42].

**Table 3.** Comparison between reconstructed $VP_{JA}$ and local historical documents.

| Year | $VP_{JA}$ Reconstruction (hpa) | Description |
|---|---|---|
| 1820 [w] | 15.55 | Floods have hit many parts of northern China. |
| 1946 [d] | 11.51 | No rain in 7 counties during spring and summer. |
| 1860 [d] | 11.32 | Plague caused by disasters hit 16 counties in Shanxi Province. |
| 1879 [w] | 15.38 | Floods have occurred in many places in Shanxi. |
| 1916 [d] | 11.88 | Drought struck many parts of Inner Mongolia. Annual rainfall is less than 250 mm. |
| 1928 [d] | 11.49 | Drought occurred throughout Inner Mongolia. The number of people affected by the drought reached more than 40,000. Food was in short supply and bandits were rampant. |
| 1929 [d] | 12.71 | The crops died from the drought. People live by eating bark and roots. Many people commit suicide because of extreme hunger. |
| 1930 [d] | 12.23 | The rainfall in Hohhot in June and August was 18.5 and 14.6 mm, respectively, significantly less than all the year round. |
| 1954 [w] | 15.78 | In summer and autumn, the whole region of Inner Mongolia is covered by heavy rain. The flooded farmland covers 133,000 hectares. The economic loss is over 100 million yuan. |
| 1987 [d] | 11.71 | Annual rainfall is 70%–90% less than normal. Ordos had almost no rain in July. Temperatures of 35–40 °C lasted for eight to nine days. |

[w] Represents the wet years. [d] Represents the dry years.

In the wet years (1819 and 1828), when the reconstruction results responded, corresponding flood records were recorded. For example, floods occurred in many areas in the 25th year of the Jiaqing of the Qing Dynasty (1820), and the court issued grain relief to ensure people's livelihood. In the seventh year of the Daoguang of the Qing Dynasty (1827), widespread floods occurred in Inner Mongolia, and in 1954, after the summer, the floods were further aggravated, with heavy rainfall in many areas. Farmlands and houses were washed down by floods, and monthly precipitation was 50–197% higher than normal precipitation [42]. In reality, however, flood disasters in Inner Mongolia tend to be localised or short-term occurrences. During 1870–1890, no significant flood disasters occurred in the Ordos region [42]. However, studies in the surrounding areas showed [24,43] that 1870–1890 were low-temperature years with high precipitation. After the late 1950s, the general trend showed drought, and in the adjacent Loess Plateau, the trends of precipitation and PDSI decreased [44–48]. This finding suggested that the weakening of the Asian summer monsoon was widely manifested in various regions after the 1950s. This change may be caused by anthropogenic aerosols [49]. A decreasing trend of the Asian summer monsoon was detected over the last 80 years by Liu et al. (2019) [38].

We compared our reconstructions with several other hydroclimatic reconstructions and found that they were significantly correlated on long time scales and showed the same long-term trends (Figure 5). For example, the reconstructed precipitation series in North China [50] showed a significant positive correlation with our reconstructions (r = 0.457, $p < 0.001$, and N = 195). The long-term variations in precipitation sequences in Shanxi region were consistent with our reconstruction (r = 0.406, $p < 0.001$, and N = 196) [33]. Moreover, our reconstructions were significantly correlated with the regional dry–wet index of the North China Plain (r = 0.387, $p < 0.001$, and N = 193) [41]. The results showed that the reconstructed sequences were significantly negatively correlated with the adjacent dry–wet sequences based on historical literature reconstruction (r = −0.411, $p < 0.001$, and N = 173) [51]. The negative correlation is obtained because high VP represented a wet

climate period, and a large index of the wet–dry series denotes a drought period. Among all the time series, two periods of high VP are similar: which were 1820s and 1950s. Since the 1950s, all the series have shown a decreasing trend (Figure 5).

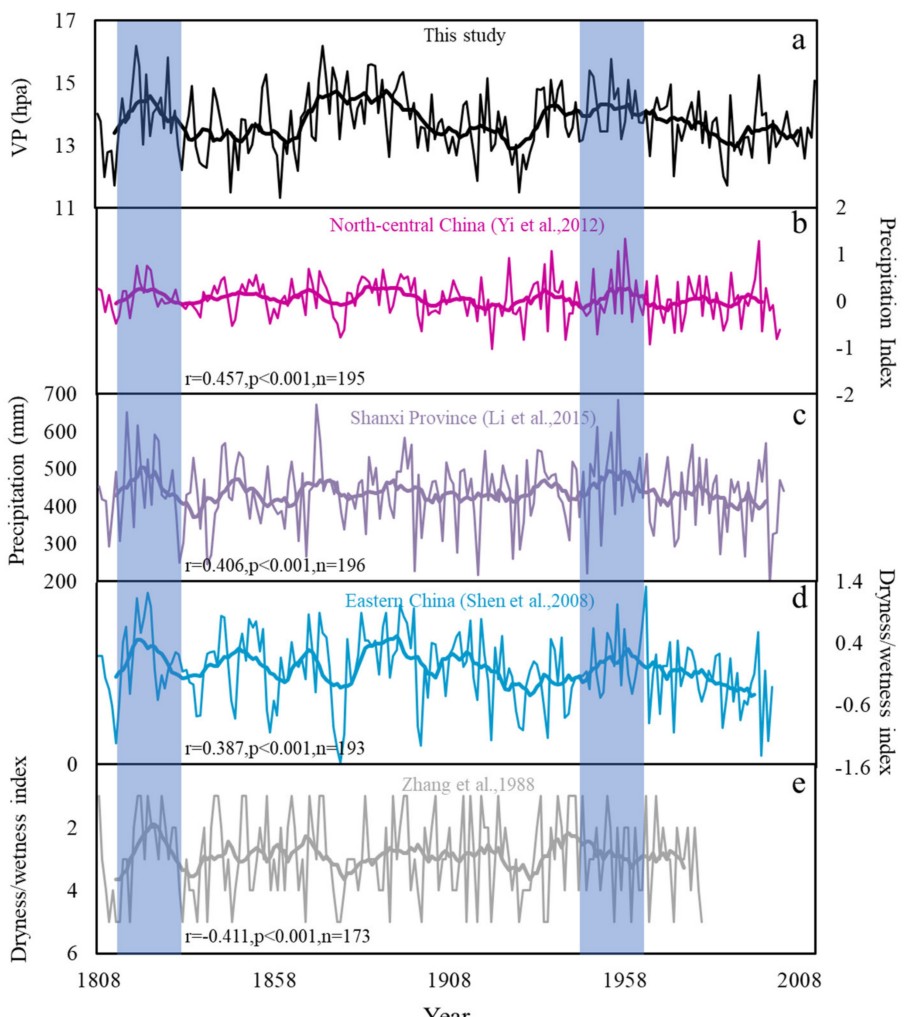

**Figure 5.** Comparison between reconstructed annual $VP_{JA}$ and existing reconstructions based on other paleoclimatic proxies. The correlation coefficients at the inter-annual time scale are shown. The thick line is an 11-year moving average. (**a**) Reconstructed annual VPJA in Ordos. (**b**) Precipitation index in North-central China reconstructed by Yi et al. 2012 [50]. (**c**) Precipitation in Shanxi province reconstructed by Li et al. 2012 [33]. (**d**) Dryness/wetness index in Eastern China reconstructed by Shen et al. 2008 [41]. (**e**) Dryness/wetness index reconstructed by Zhang et al. 1988 [51].

*4.3. Possible Factors Affecting VP Changes in the Ordos Region*

We used the EEMD method to decompose the reconstructed $VP_{JA}$ series into six IMFs and a residual trend (Figure 6), with unique characteristics reflecting fluctuations and trends on different time scales [31,52]. Among them, IMF1 and IMF2 cycles are in the range of 2–7 years, which is consistent with the El Niño–Southern Oscillation (ENSO) cycle [53,54]. The spatial correlation analysis between the IMF1 and IMF2 mean periods and sea surface temperatures (SSTs) (Figure 7) showed a significant negative correlation, suggesting a possible correlation between VP variations in the Ordos region and ENSO. Studies have shown that ENSO affects precipitation in the monsoon region through air–sea coupling [55]. Ordos is at the edge of the East Asian summer monsoon. ENSO strongly influences the East Asian monsoon, and thus, changes water resource availability [54]. Li et al. (2011) were the first researchers to find a strong correlation between stable $\delta^{18}O$ and equatorial

central-eastern Pacific SSTs in northern China tree rings [56]. The following mechanism may be used for this phenomenon: anomalous walker circulation in the EI Nino/La Nina years influenced the position of the northwest Pacific subsurface and changed the direction of the water vapour source [56]. According to the results of previous studies and our study, the effect of central Pacific SSTs on the climate of Ordos can gradually increase, further intensifying the drought.

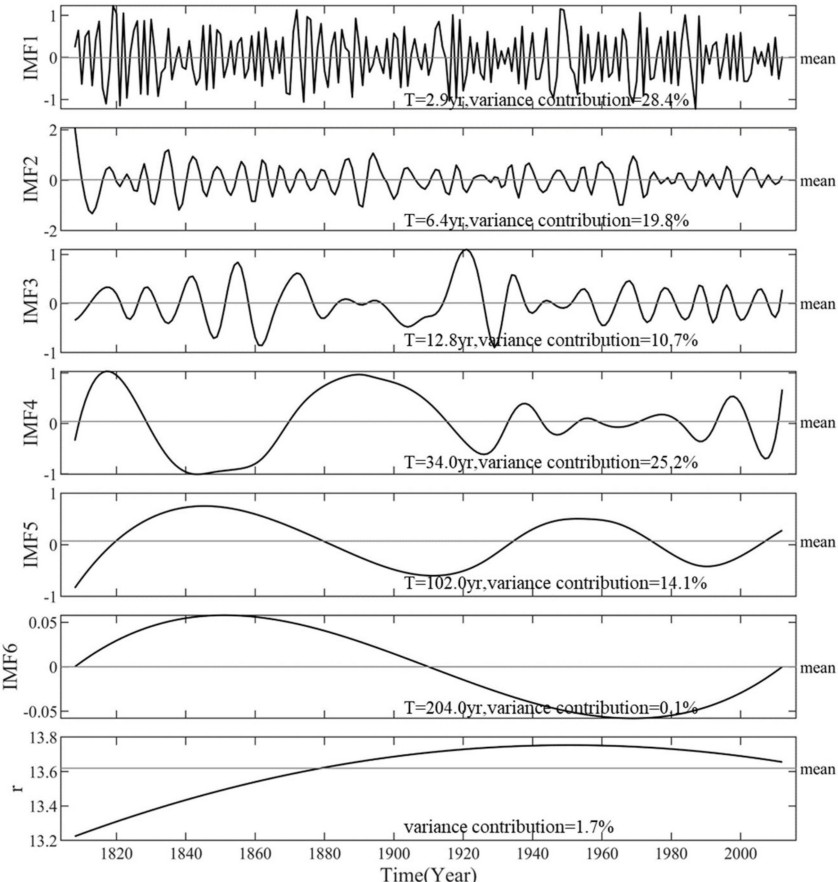

**Figure 6.** Results of the EEMD of VP$_{JA}$ reconstruction. The periodicities and variance contributions are demonstrated. IMF represents intrinsic mode function.

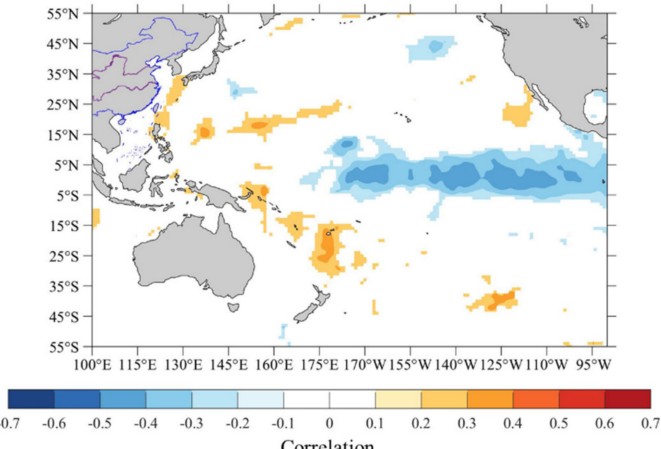

**Figure 7.** Spatial correlation between IMF1 and IMF2 average period and sea surface temperature during 1951–2012.

## 5. Conclusions

In this study, we reported the negative correlation between tree-ring cellulose $\delta^{18}O$ and VP in the Ordos region (r = $-0.688$, N = 56, and $p < 0.01$); the negative correlation is reasonable because tree-ring cellulose $\delta^{18}O$ is mainly derived from the observed $\delta^{18}O$ of precipitation. We designed a linear regression equation to reconstruct the July–August VP variations for the past 205 years. The reconstruction is regionally highly representative and shows synchronous trends with the rest of the surrounding hydroclimatic proxies. The regional VP was the highest and lowest in the 1820s and 1860s, respectively. In particular, the 1820s and 1870s exhibited a significant era of high regional VP. Since the 1950s, VP has continued to decrease. This phenomenon may result from the weakening of the Asian monsoon. The extreme VP epochs exhibited by the reconstructed sequences correspond well to the drought and flood events recorded in the historical documents. In addition, ENSO regulates VP variations and leads to changes in water availability in the East Asian monsoon region through sea–air coupling, thereby further intensifying drought in the Ordos region.

**Supplementary Materials:** The following are available online at https://www.mdpi.com/article/10.3390/f12060788/s1, Figure S1. (a) Four measured individual series and the master series of $\delta^{18}O$ produced by the numerical mixing method. (b) Mean inter-series correlation (Rbar), the running expressed population signal calculated using 30-year windows and a lag time of 15 years, Figure S2. Comparison between tree ring $\delta^{18}O$ and Global Network of Isotopes in Precipitation (GNIP) $\delta^{18}O_P$ from June to August in Baotou and Xi'an, Table S1. Statistical features of four individual series and the master series, Table S2. Correlation coefficients between each of the four individual $\delta^{18}O$ series.

**Author Contributions:** Data analysis: W.L. and Q.L.; writing original draft and revising: W.L., Q.L., H.S., R.D. and Y.L.; methodology: W.L., Q.L. and H.S.; investigation: Q.L., H.S., R.D. and Y.L.; software: W.L. and Q.L. All authors have read and agreed to the published version of the manuscript.

**Funding:** This research was funded by the Chinese Academy of Sciences (XDA23070202, XDB40000000); the National Natural Science Foundation of China (Nos. 41873021, 41991251, 32061123008 and 3191101770); the Youth Innovation Promotion Association of CAS (No. 2017451) and the Light of West China, CAS and the STEP program (Grant No. 2019QZKK0101). This work is a contribution of The Belt and Road Center of Environmental Studies, the Institute of Earth and Environment, Chinese Academy of Sciences.

**Acknowledgments:** We thank Lu Wang for the preliminary work. Data can be downloaded from the NOAA online server, https://www.ncdc.noaa.gov/paleo/study/27191 (accessed on 6 August 2020).

**Conflicts of Interest:** The authors declare no conflict of interest.

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
