# Peer review of "Cellulose δ18O of Tree Rings Reflects Vapour Pressure Variations in the Ordos Plateau"

_forests, doi:10.3390/f12060788_

Round 1

Reviewer 1 Report

(1) Rbar and EPS are determined from a table that is not in the text, at least that is how I understand the text.

(2) The authors refer to tables S1 and S2, which are not 1 and 2

(3) The label Table 2b appears in the text, which I do not see anywhere.

I still think the content is good, but the authors need to correct this inconsistency! The reason I find the content interesting is the use of stable oxygen from the trunks of the trees to determine the paleo climatological properties of the area. I believe that the relationships between stable oxygen isotopes determine the climate in much more detail than the classical ring method. I suggest that the authors correct the content and send it to me again for review!

Reviewer 2 Report

The manuscript makes a rather good impression. The language and logic are both quite clear, so, the presented results may be of interest to the readership.

Some moderate improvements to the manuscript are still required.

The caption for Fig.1 is confusing. 

(a) Monthly distributions of temperature and relative humidity - whereas (a) is the study area location

L. 136: it would be better to explain why δ18O are measured in ‰.

L. 190-191: the correlation between δ18O and precipitation do not exceed the 99% confidence level (Figure 2a). - does not? What about October?

 L. 215: Linear regression functions. - a vague sentence

Figures 4, 7: it is better to indicate what the color scale is

The Discussion is a bit lengthy. This is because that many fragments belong rather to the Results. For example, L. 326-340, 346-352, Fig. 6. There is an impression that the authors continue to tell the results.

Minor comments:

L. 131: weighed he homogeneous - the?

Table 1: VPJA - VPJA?

L. 210: temperatures mean relative humidity - comma after 'mean'

L. 358: Li et al., 2011 - no reference.

Round 2

Reviewer 1 Report

Dear authors!

Upon re-examination, I find some more ambiguities:
Line 190: Isn't relative humidity inversely correlated in September as well?
Why are July and August chosen?
I cannot understand from the text how you were able to correlate the annual values of the stable oxygen isotope ratios from tree trunks with the monthly values of precipitation, humidity, temperature?
I suggest that you make the graph of Figure 2a in color technique?

Reviewer 2 Report

The presented manuscript has been improved, so, my feeling is that it may be accepted - having in mind that it will be also subject to technical editing.
